# The First Evidence of Bacterial Foci in the Hair Part and Dermal Papilla of Scalp Hair Follicles: A Pilot Comparative Study in Alopecia Areata

**DOI:** 10.3390/ijms231911956

**Published:** 2022-10-08

**Authors:** Fabio Rinaldi, Daniela Pinto, Elisa Borsani, Stefania Castrezzati, Amedeo Amedei, Rita Rezzani

**Affiliations:** 1Human Advanced Microbiome Project-HMAP, Giuliani SpA, 20129 Milan, Italy; 2Division of Anatomy and Physiopathology, Department of Clinical and Experimental Sciences, University of Brescia, 25123 Brescia, Italy; 3Interdepartmental University Center of Research “Adaption and Regeneration of Tissues and Organs-(ARTO)”, University of Brescia, 25123 Brescia, Italy; 4Department of Experimental and Clinical Medicine, University of Florence, 50134 Florence, Italy; 5Interdisciplinary Internal Medicine Unit, Careggi University Hospital, 50134 Florence, Italy

**Keywords:** alopecia areata, microbiota, immunity, bacteria, transmission electron microscopy

## Abstract

The role of the microbiome in hair follicle (HF) growth represents a growing field of research. Here, we studied the bacterial population in the scalp hair follicles of subjects with alopecia areata (AA). Two Healthy and two AA subjects, respectively (20–60 years old), were enrolled and studied regarding the microbial community in the subepidermal scalp compartments by means of a 4-mm biopsy punch. Samples were examined by 16S sequencing, histochemical staining (Gram’s method), and transmission electron microscopy (TEM). Bacterial foci were observed in the AA subjects’ follicles with both the two adopted complementary approaches (electron microscopy and Gram staining). Significant (*p* < 0.05) differences were also found in the three-layer biopsy samples (*p* < 0.05) regarding the bacterial population. In particular, in the deep epidermis and dermis levels, a significant (*p* < 0.05) lower abundance of *Firmicutes* and a higher abundance of *Proteobacteria* were found in AA samples compared to the healthy control. *Firmicutes* also showed a significant (*p* < 0.05) lower abundance in hypodermis in AA subjects. In addition, Enterobacteriaceae and the genera *Streptococcus, Gemella, Porphyromonas,* and *Granulicatella* were relatively more abundant in AA groups at the deep epidermis level. The *Staphylococcus* and *Flavobacterium* genera were significantly less abundant in AA samples than in controls in all three-layer biopsy samples (*p* < 0.05). In contrast, *Veillonella* and *Neisseriaceae* were relatively more abundant in the healthy control group compared to the AA sample. Therefore, higher alpha diversity was observed in all three-layer biopsy samples of AA patients compared to the control. In conclusion, our data suggest that tAA could be defined as a “hair disease associated with dysregulated microbiome-immunity axis of hair follicles”.

## 1. Introduction

The alopecia areata (AA), noncicatricial alopecia, is an autoimmune disease targeting hair follicles (HF) in the anagen phase without predilection for sex or race. The AA incidence ranges from 0.57% to 3.8% [1].

The exact pathogenesis of AA remains still unknown, but increasing evidence suggests the implication of the HF immune privilege (HFIP) collapse from the bulge downwards to the bulb [2,3,4] as a possible stepping stone. However, how perifollicular memory T cells and innate lymphoid cells are implicated in HFIP collapse remains an unexplored research frontier [5]. In addition, another suggestive and intricate aspect concerns the involvement of the HFs’ microbiome in the hair cycle in AA and especially in the HFIP process [6,7]. Other autoimmune diseases, such as vitiligo, Hashimoto’s thyroiditis, psoriasis, diabetes mellitus, lupus erythematosus, and celiac disease, have been associated with comorbidities in AA patients [8]. Among these, hypothyroidism has the strongest association [9,10,11].

Thousands of microorganisms, including bacteria, fungi, archaea, and viruses, inhabit the skin and scalp ecosystem; they represent the so-called microbiota [12]. The theatre of activities and interactions with the human host of the microbiota by molecules (metabolites, DNA, genes, etc.) is called a “microbiome” [13]. There is growing evidence of the presence of bacteria below the surface of the skin and alongside the HFs [14,15,16,17]. It is also becoming increasingly clear that the dialogue between bacteria and the underlying tissue is a dynamic interplay [18,19]. It has also been hypothesized that external microbial stimulation creates immune responses in the HF epithelium [20] and, contextually, immunological events around the HF could shape microbial composition [21].

HF microenvironment, including lipid-rich and hypoxic conditions, could contribute to the establishment of resident microbes. For example, the hair shaft environment may be unfavorable to the growth of genera such as Propionibacterium, which would prefer a low level of oxygen and a high level of sebum like that of HF [22]. Indeed, exactly like the skin, the resident microbiota varies among different hair environments [14].

Recently, we reported the scalp and gut microbial shift in AA patients compared to healthy controls [23,24]. In addition, the constant crosstalk between the microorganisms living on the scalp and the skin’s immune cells is well documented [13]. Notably, the failure of the fine regulation between host immunity and “tolerance” to the resident microbiota is critical and might contribute to inflammatory and autoimmune skin diseases, including AA [25,26,27,28].

Finally, it has been suggested that the keratinocytes, in response to alteration of the resident microflora near and around HF, could unbalance the chemokines’ secretion and IP guardian, playing a role in both AA development and response to therapy [5].

Recently, we highlighted the differences in microbial populations on the scalp of AA and healthy subjects [11] and, in detail, by analyzing the Kyoto Encyclopedia of Genes and Genomes (KEGG) pathways. We documented that the pathways related to bacterial chemotaxis and cellular antigens were predominant in AA samples. The bacterial chemotaxis allows different strains to access specific host niches, delivering proapoptotic host cell factors [29]. Interestingly, microbial-derived antigens have been reported to be responsible for the susceptibility of HFs to HFIP [30].

This provides initial evidence of how our resident skin flora could interact with the immunological part of the HF ecosystem.

So, to further explore the strict relationship between microbiota changes and pathogenesis and development of AA, we performed and compared observations by electron microscopy and gram staining for in situ analysis of the spatial arrangement of bacteria around the HF of AA patients and healthy control, respectively. Finally, the identified bacterial foci were then compared with taxonomic analysis obtained using 16S rRNA analysis.

## 2. Results

We used three different methodologic approaches to observe, define, and compare the bacterial foci in scalp biopsy patients affected by alopecia areata and healthy controls.

### 2.1. Microbiota Profiling of the Scalp in AA Patients

The composition of human scalp bacterial composition of controls (*n* = 2) and AA (N = 2) subjects have been analyzed by IlluminaSeq (Figure 1). A bacterial shift was found based on three-layer biopsy samples from AA subjects. Moreover, we found significant (*p* < 0.05) differences in the microbiota composition between healthy and AA subjects based on all three-layer biopsy samples (Table 1).

Regarding the deep epidermis level and the dermis level, *Firmicutes* showed significantly lower abundances in AA samples compared to those in healthy samples. In contrast, Proteobacteria was associated with significantly higher abundances (*p* < 0.05).

Notably, at the hypodermis level, *Firmicutes* showed significantly lower abundances in AA samples compared to those in healthy samples. Finally, the *Proteobacteria* and *Bacteroidetes phyla* were relatively more abundant in AA samples than in controls (*p* < 0.05).

In addition, the family *Enterobacteriaceae* and the genera *Streptococcus*, *Gemella*, *Porphyromonas*, and *Granulicatella* were relatively more abundant in AA groups at the deep epidermis level. Therefore, *Neisseria subflava* was absent in healthy controls at the species level.

The *Staphylococcus* and *Flavobacterium* genera were significantly less abundant in AA samples than in controls in all three-layer biopsy samples (*p* < 0.05).

The genus Pseudomonas and the family *Micrococcaceae* were more abundant in the dermis and hypodermis layers of the AA subjects. In contrast, *Veillonella* and *Neisseriaceae* were relatively more abundant in the healthy control group compared to AA samples.

The Shannon index values were indicative of differences in diversity among the analyzed biopsy layers (Figure 2). In particular, in all three-layer biopsy samples of AA patients, we observed a higher alpha diversity.

### 2.2. Bacterial Infection in AA Subjects

Both the two adopted complementary approaches (electron microscopy and Gram staining) have allowed the detection of bacterial foci in the follicle of AA subjects. By the histochemical analysis, we observed that Gram-positive and negative bacteria foci were around the hair follicle (Figure 3).

### 2.3. Histology and Ultrastructure of Normal and Alopecia Aerate Scalp Hair Bulb

Before describing histologic and ultrastructural alterations in AA, the normal growth of hair follicles will be described briefly as a good starting point for explaining the morphological changes in several pathological conditions.

The hair bulb shows, from outside to inside, three major components: the epithelial hair matrix showing mainly the keratinocytes and melanocytes; the connective tissue sheath—a mesenchymal sheath wrapping the entire hair follicles; the follicular papilla—a vascularized mesenchymal structure which presents a fine basal lamina. We also recognized some keratinocytes in the epithelia hair matrix in the mitotic stage, indicating a proliferative moment linked to the hair phase (Figure 4a,b). The follicular papilla shows many fibroblasts and some fibrocytes embedded in ground substance; they are separated from the surrounding hair matrix (Figure 4b).

As previously reported, the hair follicles are altered in AA, and we confirmed, via semithin and electron microscopy sections, that these alterations involve both hair follicles and their matrix; we showed both these changes.

In this regard, by semithin sections, we showed many changes in the hair matrix, the basal lamina, and the area around the follicular papilla (Figure 5a). The basal lamina of the follicular papilla seems very thick, while the area around the papilla shows many not correctly organized cells.

Based on these results, we performed an electron microscopical analysis. We observed that the hair matrix showed many keratinocytes with cellular injuries, such as a marked thickening of their membrane that could be defined “beaded” membrane. Moreover, these cells and the other part of the hair matrix showed much apoptotic debris (Figure 5b,c).

Of note, as documented in detail in Figure 6, the follicular papilla and its surrounding area showed the presence of elongated-shaped electron-dense particles such as the bacteria infections (Figure 5c). In healthy controls, rare black particles were present, while in AA patients, the black particles were clearly present in a widespread manner. Moreover, it is important to highlight that these bacterial infections were surrounded by well-recognized lymphocytic infiltrates thanks to their rounded nucleus and their size. Further, the latter were also scattered and in the hair matrix (Figure 5c).

Finally, in addition to the bacterial infections and lymphocytic infiltrates, giant spherical melanin structures occurred in the altered follicular papilla (Figure 5c); these structures contained many melanosomes and were located between the follicular papilla at the hair matrix. Furthermore, the follicular papilla showed many fibrocytes and abundant flocculent substances concerning healthy control.

## 3. Discussion

The best and most crucial result of the study is that, for the first time, we documented the presence of bacterial foci around hair follicles in patients with alopecia areata.

We analyzed the microbiota composition considering three skin biopsy layers (deep epidermis, dermis, and hypodermis) that is in agreement with our previous works [10,11]. We confirmed the differences in microbial populations inhabiting the scalp of AA patients compared to those of healthy controls.

We observed significant differences (*p* < 0.05) in all three-layer biopsy samples, documenting an increase in some species, such as *Neisseria subflava* and *Pseudomonas*, and a decrease in *Veillonella genus* at the dermis level. We confirmed that specific microbiota characteristics are associated with damaged AA tissue. Interestingly, *Neisseria sicca/subflava*, a bacterium generally considered commensal inhabitants of the human oropharynx, was reported to cause occasional infections [31,32].

In addition, we found an increase in gram-positive cocci, such as *Micrococcaceae*. Usually, species from this family are common noninfectious bacteria of the human skin; however, recently, Boldock and colleagues [33] suggested that some pathogens might recruit them (e.g., S. aureus) to support and initiate infection.

In addition, we observed differences between different layers and confirmed that skin location is the main impacting factor on the composition of bacterial communities. Therefore, an unexpected increase in α-diversity was found in AA patients (compared to the healthy controls), suggesting a higher susceptibility of the HF to be colonized by microorganisms.

Finally, we used both optical and electron microscopy techniques to localize the microbes in the deep layers; in AA samples, the gram staining showed the presence of bacteria incorporated inside the outer root sheet at the isthmus level. Moreover, at a deep level, we identified in the follicular papilla and its surrounding area, the presence of widespread electron-dense particles such as bacterial infections (rarely observed in healthy controls). This relevant finding represents the most important result of our study. In contrast, many studies have confirmed the presence of numerous bacteria and fungi within the follicular opening and upper part of the HF (the infundibulum). Only some experimental studies have located microbes deeply in the follicle [21].In other skin pathology, such as acne vulgaris [33] and folliculitis decalvans [34], but anyone in AA. Furthermore, we found that the bacterial outbreak was surrounded by infiltrating lymphocytes suggesting an active and dysregulated role of the host immune response in this pathology.

In this scenario, it is irrelevant to remember that, like the gut microbiome, the skin microbiome has critical and crucial crosstalk with the host immune system modulating the immune responses (Figure 6) [21,35]. However, while the bacterial role in HF-associated inflammatory diseases has been well established [36], poor information is currently available on how alterations in the HF microbiota can affect the growth and immunological status of hair follicles.

Previously [11], we correlated the HF microbiota modifications, especially at the dermis level, with changes in pathways related to immune responses such as bacterial chemotaxis and glycosaminoglycan (GAG) degradation; GAGs can be used by bacteria to evade the host immunosurveillance. Therefore, we also found a positive correlation between microbial HF changes in AA patients and nucleotide-binding oligomerization domain containing 2 (NOD2), an innate immune receptor [37].

Regarding these last considerations, it is important to remember that AA is classified as an autoimmune disease. However, an increased prevalence in other forms of inflammatory skin diseases (e.g., psoriasis, vitiligo, and atopic dermatitis) were reported in AA patients, suggesting that they have an increased risk of developing T cell-driven inflammatory skin diseases [38]. Moreover, in physiological conditions, the hair follicle is defined as an “immune-privileged site” that prevents autoimmune responses [39], and the breakdown of this homeostasis is usually considered one of the major AA drivers. It has been proposed that follicular keratinocytes have the ability to release chemotactic for the T cells’ enrollment into the perifollicular area [40], thereby triggering a self-replicating process.

Additionally, the electron microscopy images showed the ultrastructural complex change in which the follicle undergoes AA pathology. In detail, we observed an altered hair follicle matrix with much apoptotic debris that affects early cortical differentiation epithelium and the presence of infiltrating lymphocytes. These data suggest that the matrix seems to be the primary target of an immune attack on the hair follicle [41,42]. In addition, these degenerative changes lead to a localized region of weakness in the hair shaft, which causes hair breaking when it emerges from the ostium at the skin surface.

Finally, we observed giant spherical melanin structures in the altered follicular papilla that is in line with previous data, showing [43,44] that the hair bulb melanocytes were damaged in acute AA, resulting in impaired melanogenesis and the loss of the ellipsoidal shape. The main limitation of the study was the small sample size. Further studies consisting of a larger number of patients are needed to support the provided evidence better.

## 4. Materials and Methods

### 4.1. Subjects’ Recruitment

The study was in the form of a monocentric study involving both healthy and AA subjects.

Healthy subjects have been screened in the absence of any history of dermatological or scalp disorders and enrolled if confirmed after clinical examination.

AA patients were diagnosed clinically and confirmed as having AA by biopsy, according to the WHO criteria. Essential clinical data were collected at baseline under dermatological control according to the guidelines of the National Alopecia Areata Foundation [45].

We adopted the following exclusion criteria: (i) the administration of antibiotics in the last 30 days before the sampling; (ii) the probiotics’ use in the last 15 days; (iii) women on pregnancy or lactation; (iv) the documented presence of other dermatological diseases; (vi) therapy with antitumor, immunosuppressant, or radiation therapy in the last 3 months; (vii) treatment with topical or hormonal therapy on the scalp in the last 3 months. The study started after the approval of the Ethical Independent Committee for Clinical, not a pharmacological investigation in Genoa (Italy). All patients were evaluated and enrolled after signing informed consent.

### 4.2. Sample Collection

Two healthy and two AA subjects, respectively (20–60 years old), were enrolled under dermatological control at the Italian private dermatological clinic, Studio Rinaldi (Milan, Italy).

The subjects were sampled using a 4-mm biopsy punch to assess the microbial community in the subepidermal scalp compartments, as previously described [10] (Figure 7). For each subject, two biopsy samples were collected from the occipital area of the scalp and, for AA subjects, in correspondence with an affected area. After collection, samples were stored in a preservative medium (Allprotect^®^ Tissue reagent, Qiagen, Milan-Italy) at +4 °C until further analysis. One biopsy sample was used for the metataxonomic profile. These samples were already analyzed in a previously published cohort [10], and sequences were deposited into the National Centre for Biotechnology Information (NCBI) BioProject database under the project number PRJNA510206. The other biopsy sample was divided into two halves, one half used for resident gram-positive and gram-negative bacteria by means of chemical staining (Gram’s method) and the other for electron microscopy.

### 4.3. DNA Extraction and 16S Amplicon Generation, Sequencing, and Analysis-Illumina Sequencing

Bacterial DNA was extracted through the QIAamp Dneasy Tissue kit (Qiagen, Milan, Italy) according to manufacturer protocol, with minor modifications [46] followed by quantification with the QIAexpert system (Qiagen, Milan, Italy) before sequencing.

Following universal prokaryotic primers were used for the V3-V4 variable region: 341 F CTGNCAGCMGCCGCGGTAA [47,48] and 806bR GGACTACNVGGGTWTCTAAT [49,50,51] at Personal Genomics (Verona, Italy) following the method of Caporaso and colleague [52] and Kozich and colleagues [53], with minor modifications. The 300PE instrument (Illumina, San Diego, CA, USA) was used for library generation.

Following bacterial 16S rRNA gene sequences, FastQC v0.11.5. was used for quality control of fastq reads and the quality trimming made by Cutadapt, v. 1.14 [54] and Sickle v. 1.33 software toolkits. OTU tables were generated by QIIME v1.9. [55]. The Greengenes database v13_8 was used as a reference for the bacterial taxonomic assignment [56]. Alpha diversity was inspected using the Shannon index.

### 4.4. Transmission Electron Microscopy (TEM)

Small skin samples obtained from the enrolled subjects were fixed in 2.5% glutaraldehyde in 0.1 M sodium cacodylate buffer, pH 7.4 (Electron Microscopy Sciences, Hatfield, PA, USA) overnight at 4 °C. They were rinsed in buffer, postfixed in osmium tetroxide in cacodylate buffer for one four at room temperature, rinsed in buffer again, and dehydrated through a graded series of acetone to 100%. Then, they were infiltrated with Araldite-Epon resin (Electron Microscopy Sciences, Hatfield, PA, USA) in a 1:1 solution of Araldite-Epon: acetone for one hour at room temperature. After, they were placed in fresh Araldite-Epon resin for one hour at 37 °C and then embedded in Araldite-Epon overnight at 60 °C. Semithin sections (1 μm) were cut on a Reichert-Jung UltraCutE ultramicrotome, collected, and stained with methylene blue. After, thin sections (70–80 nm) were cut on the same ultramicrotome, collected onto formvar-coated grids, and stained with UranyLess and lead citrate (EMS, Hatfield, PA, USA). Finally, the thin sections were observed in a transmission electron microscope (Tecnai G2 Spirit; FEI Company, Eindhoven, the Netherlands) at 80 kV. Images were collected using a digital imaging system. Two blinded observers evaluated 10 images for each sample according to previous work for morphology and the presence of electron-dense particles ascribable to bacteria or lymphocytes [42,57].

### 4.5. Tissue Gram Staining

Cryosections (6 μm) were fixed in Acetone and stained for Gram stain (Sigma-Aldrich S.r.l., Milan, Italy) by procedures described by the manufacturer.

### 4.6. Statistical Analysis

Statistically significant differences in bacterial community determined by AA condition were determined by Student’s t with Welch’s correction. Analyses were performed with GraphPad Prism 7.0 (GraphPad Software, Inc., San Diego, CA, USA). *p*–values equal to or less than 0.05 were considered significant.

## 5. Conclusions

In conclusion, our data suggest that the alopecia areata is not a single clinic-pathological entity. However, it has to be managed in a holistic way considering its complexity and the alterations of microbiome homeostasis, so that it could be defined as a “hair disease associated with dysregulated microbiome-immunity axis of hair follicles”.

## Figures and Tables

**Figure 1 ijms-23-11956-f001:**
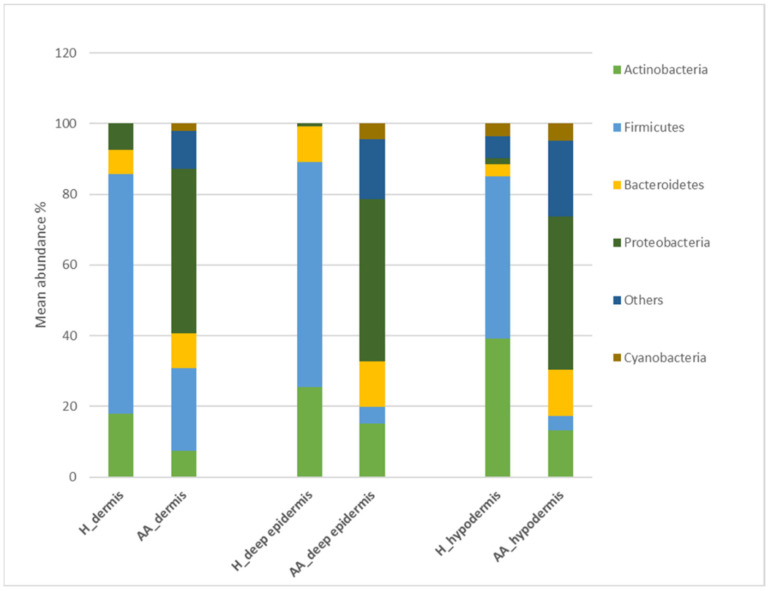
Bacterial profiling in control and AA subjects. Percentage of bacteria at phylum level in the healthy controls (H) and patients with alopecia areata (AA) groups. Results are presented as the percentage (%) of total sequences.

**Figure 2 ijms-23-11956-f002:**
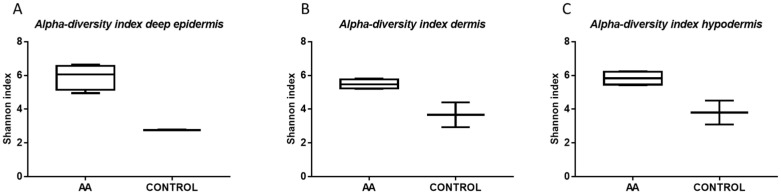
Alpha diversity index. All the diversity indices for (**A**) deep epidermis, (**B**) dermis, and (**C**) hypodermis in AA and healthy control group are expressed as Shannon index.

**Figure 3 ijms-23-11956-f003:**
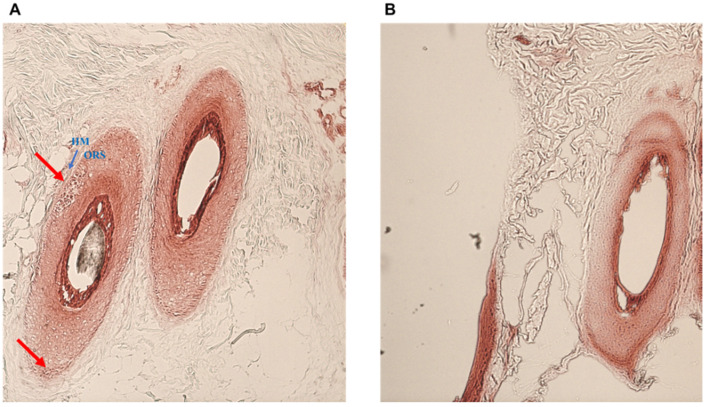
Gram staining. Hair follicles of AA patients (**A**) and control subjects (**B**) at the level of the isthmus showing bacterial foci (red arrow). HM: hyaline membrane (blue arrow); ORS: outer root sheet.

**Figure 4 ijms-23-11956-f004:**
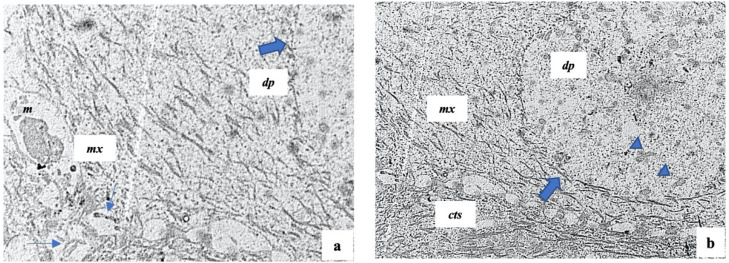
Representative electron microscopical overview of normal scalp hair follicles. *cts*: connective tissue sheath, *mx*: hair matrix; *dp*: follicular papilla; *m*: melanosomes; thin arrows: keratinocytes in mitotic stage; thick arrows: fine basal lamina of follicular papilla; arrowheads: fibroblasts. (**a**) 6200×; (**b**) 6200×.

**Figure 5 ijms-23-11956-f005:**
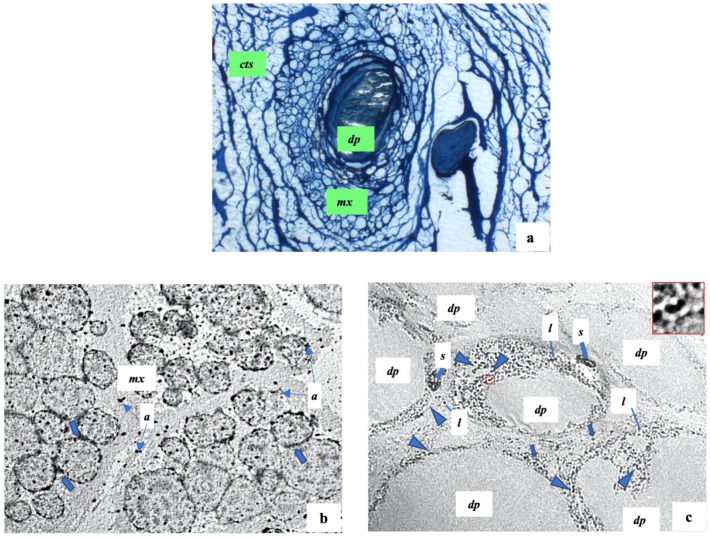
Scalp hair follicles in alopecia areata. (**a**) A representative semithin section shows: a follicular papilla (*dp*) with abundant flocculent substance; a hair matrix (*mx*) with many cells not well organized; *cts*: connective tissue sheath. (**b**,**c**) The representative ultrastructural sections of hair matrix and follicular papilla respectively show: *mx*: hair matrix; *a*: apoptotic debris; thick arrows: beaded membrane of keratinocytes; *dp*: follicular papilla; head arrows: bacteria infiltrations; *s*: melanosome structures; *l*: lymphocytes; red square: area of inset magnification. (**a**) 200×; (**b**) 1750×; (**c**) 12,500×.

**Figure 6 ijms-23-11956-f006:**
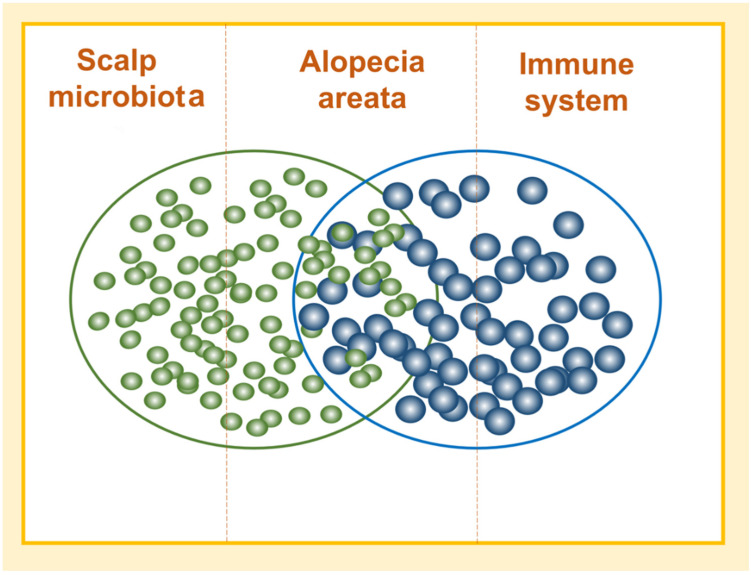
Microbiota and immunity crosstalk in alopecia areata.

**Figure 7 ijms-23-11956-f007:**
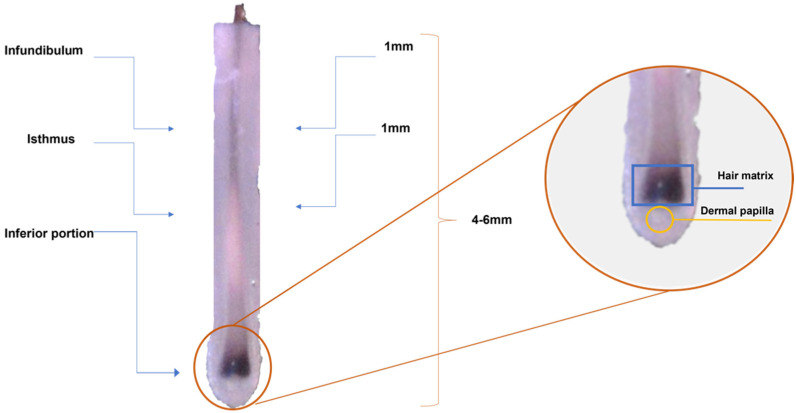
Stereomicroscopic view of the anatomy of an anagen hair follicle.

**Table 1 ijms-23-11956-t001:** Statistically different taxa emerged between microbial taxa healthy abundances versus alopecia areata (AA) at sub-epidermal layers (deep epidermis, dermis, hypodermis).

Layer	Taxa Level	Feature	Healthy	AA	*p*-Value
deep epidermis	family	Enterobacteriaceae	0.00	17.09	0.500
deep epidermis	genus	Streptococcus	2.45	16.30	0.551
deep epidermis	genus	Gemella	0.00	2.40	0.500
deep epidermis	genus	Porphyromonas	0.00	2.14	0.500
deep epidermis	genus	Granulicatella	0.00	2.50	0.500
deep epidermis	genus	Staphylococcus	49.60	0.00	0.227
deep epidermis	genus	Flavobacterium	8.00	0.00	0.156
dermis	phylum	Firmicutes	71.92	22.50	0.009
dermis	phylum	Proteobacteria	7.48	46.55	0.027
dermis	family	Micrococcaceae	0.00	6.41	0.137
dermis	genus	Staphylococcus	48.35	1.35	0.278
dermis	genus	Flavobacterium	5.00	0.00	0.275
hypodermis	phylum	Firmicutes	45.75	4.05	0.033
hypodermis	phylum	Proteobacteria	1.8	43.35	0.009
hypodermis	family	Bacteroidetes	2.96	12.74	0.015
hypodermis	family	Micrococcaceae	0.00	4.85	0.06
hypodermis	family	Neisseriaceae	0.00	1.00	0.500
hypodermis	genus	Staphylococcus	10.20	0.90	0.021
hypodermis	genus	Flavobacterium	7.10	0.00	0.269
hypodermis	genus	Veillonella	0.00	4.30	0.500

## Data Availability

The datasets used during and/or analyzed during the current study are available from the corresponding author upon reasonable request.

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
