# Peer review of "The First Evidence of Bacterial Foci in the Hair Part and Dermal Papilla of Scalp Hair Follicles: A Pilot Comparative Study in Alopecia Areata"

_ijms, 2022, doi:10.3390/ijms231911956_

Round 1

Reviewer 1 Report

The manuscript "The first evidence of bacterial foci in the anaerobe part of scalp hair follicles: a pilot comparative study in alopecia areata" contains an interesting observation that can be significant. However, the data is preliminary, and the limited sample size does not justify having a wider audience. The authors have barely touched on the importance of autoimmunity. The role the thyroid plays in AA is wholly ignored. The manuscript requires an increased sample size, comprehensive reference literature, and a detailed introduction. The revised manuscript with more data can add to our knowledge and understanding of AA.

Author Response

Dear Editor,

We revised the paper entitled “The first evidence of bacterial foci in the hair part and dermal papilla of scalp hair follicles: a pilot comparative study in alopecia areata.” submitted to ‘International Journal of Molecular Sciences (ijms-1835855).

We wish to thank you the Editor and the Reviewers for their time and expertise. We have attempted to address each concern with either additional information and/or clarifications within the text. The critiques proved very helpful, and we feel that in addressing these concerns, we have improved the quality of our study. Below, we handle each critique and specifically detail where in the revised manuscript we addressed the point. Moreover, we marked in red color the corrections in the revised manuscript.

REPLY to Reviewer 1

The manuscript "The first evidence of bacterial foci in the anaerobe part of scalp hair follicles: a pilot comparative study in alopecia areata" contains an interesting observation that can be significant. However, the data is preliminary, and the limited sample size does not justify having a wider audience.

The authors have barely touched on the importance of autoimmunity. The role the thyroid plays in AA is wholly ignored. The manuscript requires an increased sample size, comprehensive reference literature, and a detailed introduction. The revised manuscript with more data can add to our knowledge and understanding of AA.

REPLY: Thank you for your precious observation. We changed the nature of the article on page 1 line 1: “Article Case Report”.

Moreover, the following paragraph has been added to the INTRODUCTION (page 1, lines 50-53).

“Other autoimmune diseases such vitiligo, Hashimoto's thyroiditis, psoriasis, diabetes mellitus, lupus erythematosus and celiac disease, have been associated as comorbidities in AA patients [8]. Among these, hypothyroidism has the strongest association [9-11].”

We added the following references:

Naik PP, Farrukh SN. Association between alopecia areata and thyroid dysfunction. Postgrad Med. 2021 Nov;133(8):895-898. doi: 10.1080/00325481.2021.1974689. Epub 2021 Sep 6. PMID: 34455910.

Puavilai S, Puavilai G, Charuwichitratana S, et al. Prevalence of thyroid diseases in patients with alopecia areata. Int J Dermatol. 1994;33:632–633.7.

Lewinski A, Broniarczyk-Dyla G, Sewerynek E, et al. Abnormalities in structure and function of the thyroid gland in patients with alopecia areata. J Am Acad Dermatol. 1990;23(4 Pt 1):768–769.8.

Thomas EA, Kadyan RS. Alopecia areata and autoimmunity: a clinical study. Indian J Dermatol. 2008;53:70–74.

We also revised the introduction with more details.

See lines: 57-67, page 2

“There is growing evidence of the presence of bacteria below the surface of the skin and alongside the HFs [14-17]. It is also becoming increasingly clear that the dialogue between bacteria and the underlying tissue is a dynamic interplay [18,19]. It has also been hypothesized that external microbial stimulation creates immune responses in the HF epithelium [20] and, contextually, immunological events around the HF could shape microbial composition [21].

HF microenvironment, including lipid-rich and hypoxic conditions, could contribute to the establishment of resident microbes. As an example, the hair shaft environment may be unfavorable to the growth of genera such as Propionibacterium who would prefer a low level of oxygen and a high level of sebum like that of HF [22]. Indeed, exactly like the skin, the resident microbiota varies among different hair environments [14].”

References added:

Lousada MB, Lachnit T, Edelkamp J, Rouillé T, Ajdic D, Uchida Y, Di Nardo A, Bosch TCG, Paus R. Exploring the human hair follicle microbiome. Br J Dermatol. 2021 May;184(5):802-815. doi: 10.1111/bjd.19461. Epub 2021 Feb 18. PMID: 32762039.

Constantinou A, Kanti V, Polak-Witka K, Blume-Peytavi U, Spyrou GM, Vogt A. The Potential Relevance of the Microbiome to Hair Physiology and Regeneration: The Emerging Role of Metagenomics. Biomedicines. 2021 Feb 26;9(3):236. doi: 10.3390/biomedicines9030236. PMID: 33652789; PMCID: PMC7996884.

Ho BS, Ho EXP, Chu CW, Ramasamy S, Bigliardi-Qi M, de Sessions PF, Bigliardi PL. Microbiome in the hair follicle of androgenetic alopecia patients. PLoS One. 2019 May 3;14(5):e0216330. doi: 10.1371/journal.pone.0216330. PMID: 31050675; PMCID: PMC6499469.

Zheng, D., Liwinski, T. & Elinav, E. Interaction between microbiota and immunity in health and disease. Cell Res 30, 492–506 (2020). doi: org/10.1038/s41422-020-0332-7

Racine PJ, Janvier X, Clabaut M, Catovic C, Souak D, Boukerb AM, Groboillot A, Konto-Ghiorghi Y, Duclairoir-Poc C, Lesouhaitier O, Orange N, Chevalier S, Feuilloley MGJ. Dialog between skin and its microbiota: Emergence of "Cutaneous Bacterial Endocrinology". Exp Dermatol. 2020 Sep;29(9):790-800. doi: 10.1111/exd.14158.

Kabashima K., Honda T., Ginhoux F., Egawa G. The immunological anatomy of the skin. Nat. Rev. Immunol. 2019;19:19–30. doi: 10.1038/s41577-018-0084-5.

Belkaid, Y.; Segre, J.A. Dialogue between skin microbiota and immunity. Science 2014, 346, 954–959. doi: 10.1126/science.1260144.

Sanford J.A., O’Neill A.M., Zouboulis C.C., Gallo R.L. Short-Chain Fatty Acids from Cutibacterium acnesActivate Both a Canonical and Epigenetic Inflammatory Response in Human Sebocytes. J. Immunol. 2019;202:1767–1776. doi: 10.4049/jimmunol.1800893.

As regards sample size we highlighted this aspect in the limitations section:

See line 262-263, page 8 “The main limitation of the study was the small sample size. Further studies consisting of a larger number of patients are needed to better support the provided evidence.”

Usually, AA subjects undergone to a biopsy for histopathological examination. It’s very difficult for the clinician to  convincing the patient for two biopsies and to limit the morbidity, the clinicians started doing a single biopsy. Therefore, obtain biopsy samples from healthy subjects is particularly difficult especially for ethical issues.

Best regards,

Rita Rezzani

Reviewer 2 Report

The authors have presented a very important work on the first detection of bacterial foci in the anaerobic part of scalp hair follicles.

However, there are lots of concerns that should be addressed first:

1. nature of the manuscript: My major concern is the number of individuals involved in this study. It is more of a "case report" rather than a research article. The sample of two people is far too small.

2. abstract:

(a) the authors wrote "40% male"? what does that mean? this percentage is unnecessary. delete it, or give the whole numbers, i.e. 1/4

b) the title refers to the "anaerobic part", so this should also be highlighted in the abstract (next to ... "subepidermal scalp compartments"...line 20)

c) no bacterial profile (genera or families) was given in the summary. please insert the results.

3. introduction:

(a) The introduction section should include a more detailed description of the anaerobic part of the scalp, as this is the main topic (it is in the title).

b) all bacteria should be written in italics.

(c) percentages should be deleted or converted to whole numbers as in the abstract.

4. results:

(a) The authors wrote about the bacterial foci in the anaerobic part, but not many obligate anaerobic bacteria were found, except Porphyromonas.

Is there a reason for this in the technical part of the study?

b) This should be discussed in the discussion section.

c) The authors worked with the electron microscope, but why did not they take pictures showing the bacteria in more detail to see cocci or bacilli? from these pictures it is very difficult to see if it is bacterial or lymphocytic infiltration or something else

d) Figure 1: This figure should have a more detailed bacterial profile as genus or family. This figure does not follow the text from lines 88-96.

e) Figure 3: the bacteria in the figure are more brown then Gram-stained (blue cocci as micrococci and staphylococci and red as Porphyromonas or red bacilli as enterobacteria and Pseudomonas ). in the picture it is very hard to see the exact colour and cocci shape. I can see that they look more like bacilli? And where are bacilli, since the authors specify only cocci in the picture description? Another type of staining would be more acceptable, or more specific, like immunofluorescence.

f) Figures 4 and 5: Please enlarge the letters in the figures to make them more visible.

g) The statement about lymphocytes is in Figure 5c in line 164: It is very difficult to clarify whether it is bacteria or just lymphocytes. The bacteria themselves, as I said, are difficult to see in pictures.

5. Discussion:

LIne 199:

The authors stated that there is lymphocyte infiltration around the bacteria. However, this is not certain. It should be confirmed with immunohistochemistry or otherwise (a more specific method). This statement can be found in Figure 5c, line 164 as well or else. It is very difficult to clarify whether it is bacteria or just lymphocytes. this statement should be changed.

Author Response

Dear Editor,

We revised the paper entitled “The first evidence of bacterial foci in the hair part and dermal papilla of scalp hair follicles: a pilot comparative study in alopecia areata.” submitted to ‘International Journal of Molecular Sciences (ijms-1835855).

We wish to thank you the Editor and the Reviewers for their time and expertise. We have attempted to address each concern with either additional information and/or clarifications within the text. The critiques proved very helpful, and we feel that in addressing these concerns, we have improved the quality of our study. Below, we handle each critique and specifically detail where in the revised manuscript we addressed the point. Moreover, we marked in red color the corrections in the revised manuscript.

REPLY to Reviewer 2

The authors have presented a very important work on the first detection of bacterial foci in the anaerobic part of scalp hair follicles.

However, there are lots of concerns that should be addressed first:

  1. nature of the manuscript: My major concern is the number of individuals involved in this study. It is more of a "case report" rather than a research article. The sample of two people is far too small.

REPLY: Thank you for your suggestion. We agree to change the nature of the article in “Case Report”. We changed the nature of the article on page 1 line 1: “Article Case Report”.

  1. abstract:

(a) the authors wrote "40% male"? what does that mean? this percentage is unnecessary. delete it, or give the whole numbers, i.e. 1/4

REPLY: Thank you for your precious suggestion. The words “…; 40% male).” (Abstract, page 1, line 20; Materials and Methods, page 9, line 283) have been deleted.

  1. b) the title refers to the "anaerobic part", so this should also be highlighted in the abstract (next to ... "subepidermal scalp compartments"...line 20)

REPLY: Thank you for your precious suggestion. The title has been changed as follows: “The first evidence of bacterial foci in the hair matrix and dermal papilla of scalp hair follicles: a pilot comparative study in alopecia areata”.

  1. c) no bacterial profile (genera or families) was given in the summary. please insert the results.

REPLY: Thank you for your observation. So, we added in the Abstract the following sentence: “In particular, in the deep epidermis and dermis level, a significant (p<0.05) lower abundance of Firmicutes and higher abundance of Proteobacteria were found in AA samples compared to healthy control. Firmicutes also showed a significant (p<0.05) lower abundance in hypodermis in AA subjects.  In addition, Enterobacteriaceae and the genera Streptococcus, Gemella, Porphyromonas and Granu-licatella were relatively more abundant in AA groups at the deep epidermis level.

The Staphylococcus and Flavobacterium genera were significantly less abundant in AA samples than in controls in all three-layer biopsy samples (p<0.05).

In contrast, Veillonella and Neisseriaceae were relatively more abundant in the healthy control group compared to AA sample. Therefore, and a higher alpha-diversity was observed in all three-layer biopsy samples of AA patients, compared to control” (Abstract, page 1, line 26-37).

  1. introduction:

(a) The introduction section should include a more detailed description of the anaerobic part of the scalp, as this is the main topic (it is in the title).

REPLY: Thank you again for this consideration. We added the following sentences:

“There is growing evidence of the presence of bacteria below the surface of the skin and alongside the HFs [14-17]. It is also becoming increasingly clear that the dialogue between bacteria and the underlying tissue is a dynamic interplay [18,19]. It has also been hypothesized that external microbial stimulation creates immune responses in the HF epithelium [20] and, contextually, immunological events around the HF could shape microbial composition [21].

HF microenvironment, including lipid-rich and hypoxic conditions, could contribute to the establishment of resident microbes. As an example, the hair shaft environment may be unfavorable to the growth of genera such as Propionibacterium who would prefer a low level of oxygen and a high level of sebum like that of HF [22]. Indeed, exactly as the skin, the resident microbiota varies among different hair environments [14].” (Introduction, page 2, lines 57-67).

  1. b) all bacteria should be written in italics.

REPLY: Thank you for this observation. All bacteria have been written in Italics; the changes have been marked in red color throughout the text.

(c) percentages should be deleted or converted to whole numbers as in the abstract.

REPLY: Thank you for your suggestion. The percentages reported in the following sentence “The AA incidence ranges from 0.57% to 3.8%.” are detailed in the cited review article [1].

  1. results:

(a) The authors wrote about the bacterial foci in the anaerobic part, but not many obligate anaerobic bacteria were found, except Porphyromonas. Is there a reason for this in the technical part of the study?

  1. b) This should be discussed in the discussion section.

REPLY: Thank you for your observation. We revised the manuscript accordingly and changed the title to “The first evidence of bacterial foci in the hair matrix and dermal papilla of scalp hair follicles: a pilot comparative study in alopecia areata”

Through these changes, we enlarged the concept to both aerobic and anaerobic parts.

  1. c) The authors worked with the electron microscope, but why did not they take pictures showing the bacteria in more detail to see cocci or bacilli? from these pictures it is very difficult to see if it is bacterial or lymphocytic infiltration or something else

REPLY: Thank you for your suggestion. We added a magnification inset in figure 5 showing the bacteria shape (page 5). The legend of the figure has been updated as follows: “ ….l: lymphocytes; red square: area of inset magnification. a) 200x;…” (Results, page 5, line 165).

  1. d) Figure 1: This figure should have a more detailed bacterial profile as genus or family. This figure does not follow the text from lines 88-96.

REPLY: Thank you for your observation. We added table 1 (line 102 and 122-126, page 3). 

  1. e) Figure 3: the bacteria in the figure are more brown then Gram-stained (blue cocci as micrococci and staphylococci and red as Porphyromonas or red bacilli as enterobacteria and Pseudomonas ). in the picture it is very hard to see the exact colour and cocci shape. I can see that they look more like bacilli? And where are bacilli, since the authors specify only cocci in the picture description? Another type of staining would be more acceptable, or more specific, like immunofluorescence.

See lines 137-139: “Figure 3. Gram staining. Hair follicle of AA patients (A) and control subjects (B) at the level of the isthmus showing bacterial foci cocci (red arrow). HM: hyaline membrane (blue arrow); ORS: outer root sheet.”

See lines 213-214: “Finally, we used both optical and electron microscopy techniques to localize the microbes in the deep layers, and in AA samples, the gram staining showed the presence of cocci bacteria incorporated inside the outer root sheet at the isthmus level.”

REPLY: Thank you for your observation. According to your suggestions, we analyzed the images again, with a powerful tool, and didn’t notice cocci but bacteria look more than bacilli. We corrected the figure description and text accordingly.

  1. f) Figures 4 and 5: Please enlarge the letters in the figures to make them more visible.

REPLY: Thank you for your suggestion. The letters in figures 4 and 5 have been enlarged and the figures have been substituted (Results, pages 4-5).

  1. g) The statement about lymphocytes is in Figure 5c in line 164: It is very difficult to clarify whether it is bacteria or just lymphocytes. The bacteria themselves, as I said, are difficult to see in pictures.

REPLY: Thank you again for your suggestion. As written before, we added a magnification inset in figure 5 showing the bacteria shape (page 5). The legend of the figure has been updated as follows: “ ….l: lymphocytes; red square: area of inset magnification. a) 200x;…” (Results, page 5, line 190).

  1. Discussion:

Line 199: The authors stated that there is lymphocyte infiltration around the bacteria. However, this is not certain. It should be confirmed with immunohistochemistry or otherwise (a more specific method). This statement can be found in Figure 5c, line 164 as well or else. It is very difficult to

clarify whether it is bacteria or just lymphocytes. This statement should be changed.

REPLY: Thank you again for this observation. The statements are supported by the TEM images and the detailed related literature. We modified the sentence (line 331-334, page 10) as follow: “Two blinded observers evaluated 10 images for each sample according to previous work for morphology and the presence of electron-dense particles ascribable to bacteria or lymphocytes [58,59]”.

Best regards,

Rita Rezzani

Round 2

Reviewer 1 Report

The authors have answered all the questions to the best of their abilities. I have no major concerns; the manuscript is interesting. 

Reviewer 2 Report

The authors made all necessary changes.